# QCR: Quantised Codebooks for Retrieval

## Abstract

In recent years, the application of language models (LMs) to retrieval tasks has gained significant attention. Dense retrieval methods, which represent queries and document chunks as vectors, have gained popularity, but their use at scale can be challenging. These models can under-perform traditional sparse approaches, like BM25, in some demanding settings, e.g. at web-scale or out-of-domain. Moreover the computational requirements, even with approximate nearest neighbour indices (ANN) can be hefty. Sparse methods, remain, thanks to their efficiency, ubiquitous in applications. In this work, we ask whether LMs can be leveraged to bridge this gap. We introduce Quantised Codebooks for Retrieval (QCR): we encode queries and documents as bags of latent discrete tokens, learned purely through a contrastive objective. QCR's encodings can be used as a drop-in replacement for the original string in sparse retrieval indices, or can be instead used to complement the text with higher-level semantic features. Experimental results demonstrate that QCR outperforms BM25 with vanilla text on the challenging MSMARCO dataset. What is more, when used in conjunction with standard lexical matching, our representation yield and absolute 15.6% gain over BM25's Success@100, highlighting the complementary nature of textual and learned discrete features.

## 1 Introduction

Information retrieval (IR) systems have long relied on traditional term matching methods, representing both queries and documents as bags of words, i.e. very high-dimensional, sparse vectors, whose non-zero components are based on lexical frequencies estimated on text corpora. Approaches such as BM25 (Robertson & Zaragoza, 2009; Robertson & Walker, 1997) have withstood the test of time because of their effectiveness, but also in no small part due to its simplicity and scalability. This scalability is a product of the sparsity of the representations, that allows the use of efficient and flexible data structures for search, such as the inverted index (Knuth, 1997). However, traditional sparse retrieval methods are now losing their dominance. Since they are based on exact or near-exact term matching between queries and documents, they face the well-known vocabulary mismatch problem (Berger et al., 2000). Vocabulary mismatch arises when concepts with same meaning are worded differently in the query and the related document, causing sparse models to fail at the retrieval task, — e.g. with the synonyms *pants* and *trousers*.

In recent years, so-called dense retrieval models have become widely used in text applications (Wang et al., 2024; Lee et al., 2019). Embedding models typically encode queries and documents as continuous, fixed-length vectors, computing relevance scores with dot product. Powered by the recent improvements in language model pre-training, dense approaches have shown substantial improvements in IR tasks (Karpukhin et al., 2020; Xiong et al., 2020; Izacard et al., 2022; Wang et al., 2024), particularly in scenarios where large annotated datasets are available. Dense models can capture the higher-level meaning of queries and documents, enabling retrieval to go beyond simple term matching. Despite their success, dense retrieval methods come with significant drawbacks, including high computational and memory costs (Cao et al., 2021), and the need for complex approximate nearest neighbor indices (Douze et al., 2024; Malkov & Yashunin, 2018). Moreover, dense retrieval systems can underperform simple term-matching where large-scale supervised data is unavailable or difficult to collect (Ni et al., 2022).

In this paper, we introduce a method that improves standard term-matching retrieval performance by leveraging learnt, discrete embeddings of queries and documents composed of tokens from a latent codebook. These latent encodings can be used in place of natural language words with no

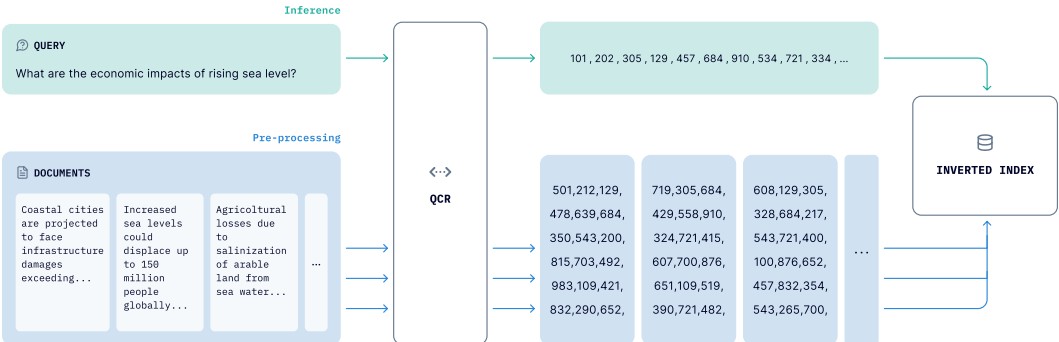

Figure 1: High-level structure of QCR. After training, we process the corpus to generate a discrete representation for every document (or passage). These discrete encodings are used to create an inverted index. At inference time, we encode an upcoming query and match it against our collection using standard sparse retrieval methods.

modification to the existing indices used in sparse retrieval methods like BM25. Borrowing from the field of discrete representation learning, we apply Finite Scalar Quantization (FSQ) (Mentzer et al., 2023), a recently introduced vector quantisation method, on top of BERT (Devlin et al., 2019) embeddings to generate discrete latent codes, which are optimised through backpropagation using a ColBERT-like loss function (Khattab & Zaharia, 2020). We hypothesise the these discrete codes can mitigate the vocabulary mismatch problem as they incorporate learnt, semantic information into their discrete representation while retraining most of the computational benefits of sparse methods.

We empirically demonstrate that our discrete representations improve BM25 performance when used in isolation and that further gains are achieved when combining these representations with the original textual queries and documents. This finding suggests that discrete representations and text provide complementary signals that enhance retrieval effectiveness when used together. This hybrid approach allows us to combine the interpretability and computational efficiency of traditional sparse methods with the semantic information of learned representations, offering a lightweight alternative to dense retrieval methods, making it particularly well-suited for large-scale applications where dense retrieval might be costly.

While our method does not outperform state-of-the-art dense retrieval models in terms of raw retrieval metrics, it offers a much more computationally efficient solution, making it highly attractive in scenarios where dense retrieval methods are impractical due to their resource demands.

Our contributions are as follows:

1. We propose a method for generating discrete representations of queries and documents, which can be used both as standalone inputs for inverted-index retrieval systems and in conjunction with traditional lexical representations.

2. Our approach improves retrieval performance on the challenging MSMARCO benchmark, demonstrating its effectiveness in both supervised and unsupervised settings. Notably, it outperforms BM25 on all reported metrics when using discrete representations alone and shows further gains when combining them with original text.

3. We highlight the practical advantages of integrating discrete representations with lexical retrieval, offering a computationally efficient alternative to dense retrieval methods while maintaining strong performance.

## 2 BACKGROUND

In Figure 1 we provide a schematic architecture of our QCR method. Our method relies on the recently introduced FSQ quantization method (Mentzer et al., 2023) to induce a discretisation of the Transformer model's output hidden states, while optimising end-to-end with a ColBERT-like loss. In the rest of this section, we provide the background useful understand the main technical contribution of our work.

## 2.1 COLBERT

ColBERT (Contextualized Late Interaction over BERT) (Khattab & Zaharia, 2020) is an information retrieval model that combines the strengths of both traditional sparse retrieval techniques and dense vector-based retrieval. At its core, ColBERT leverages deep contextual embeddings generated by BERT to represent both queries and documents. However, instead of embedding into a single fixed-length vector like most dense retrieval models, ColBERT retains individual token-level embeddings, thus preserving fine-grained contextual information. Like in term-based sparse retrieval approaches, in ColBERT the relevance score is computed as the max-pooling over the dot product between individual query and document terms.

The ColBERT scoring is given by the following equation:

$$\text{score}_{\text{ColBERT}}(q, d) := \sum_{q_i \in q} \max_{d_j \in d} \frac{e_{q_i} \cdot e_{d_j}^T}{\| e_{q_i} \| \cdot \| e_{d_j} \|} \tag{1}$$

where $e_{q_i} \in E_Q$ and $e_{d_i} \in E_D$ are the embeddings of query and document tokens, respectively.

## 2.2 FSQ

FSQ (Mentzer et al., 2023) is a technique used to discretise continuous data into a finite set of values, which is particularly useful in scenarios where memory efficiency and computational simplicity are priorities. FSQ maps each continuous value to the nearest point in a predefined set of quantised levels, in contrast with its precursor VQ-VAE (van den Oord et al., 2018), where the codebook is learnt and actively changes during training.

FSQ works by rounding each entry in the latent representation $z$ to the nearest integer, usually after applying a bounding function such as the hyperbolic tangent. To propagate gradients through the rounding operation, it simply uses straight-through estimation (STE) (Bengio et al., 2013). FSQ has been shown to be easier to optimise than other similar methods, while avoiding the issue of codebook under-utilisation that VQ-VAE suffers from.

FSQ obtains a discrete representation of a continuous vector $z$ by applying the following operation on each one of its dimensions:

$$z_i = \text{round}(\lfloor L/2 \rfloor \tanh \left( z_i^{\text{prequant}} \right)) \tag{2}$$

Here, $z_i^{\text{prequant}}$ represents the pre-quantized value of the $i$-th dimension and $L$ is the number of discrete values of each dimension. Considering the same value $L$ across all dimensions, and a total number of $d$ dimensions ($|z_i| = d$), the dimension of the codebook is equal to $|\mathcal{C}| = L^d$.

## 2.3 DISCRETE REPRESENTATION LEARNING

Discrete representations, particularly through methods such as VQ-VAE and FSQ, have gained traction in generative modeling, especially in the domain of image generation (Razavi et al., 2019; Ramesh et al., 2021; Gu et al., 2022; Chang et al., 2022), but also music (Dhariwal et al., 2020) and video (Rakhimov et al., 2020). These techniques enable the transformation of continuous data (e.g., pixel values) into discrete codes, which can be used to produce high-quality images. Discretisation methods have also been applied to text generation (Zhang et al., 2024) to allow for better control of textual output in LLMs.

Perhaps most similar to our method, Sun et al. (2023) propose GENRET to learn discrete document identifiers for the purpose of using them during generative retrieval. Their goal is to obtain identifiers that contain semantic information about the document and are easier for the LLM to generate. They use an auto-encoding framework to generate docids that are autoregressive in nature and diverse enough to avoid duplicate docids across multiple documents. Similarly, Wang et al. (2023) propose a hierarchical k-means algorithm to generate docids for each documents.

## 3 METHODOLOGY

Our proposed approach builds on the idea of leveraging LLMs to generate discrete representations optimized for retrieval tasks. We aim to combine the semantic richness of dense embeddings with

the efficiency and interpretability of sparse methods. Below, we outline the key components of our method, which includes FSQ and a novel loss function optimized for information retrieval.

### 3.1 ARCHITECTURE

We use an encoder LLM architecture, BERT (Devlin et al., 2019) specifically, to obtain embedding representations for each query and document. In particular, we augment the input text, both for queries and documents, by prepending a set of special tokens to each text string. These special tokens serve as unique markers that define fixed positions for the embedding extraction.

Given a text input, we tokenize it, add special tokens, and then feed the modified text into a pre-trained BERT model. In other words, for a tokenised query $q = \langle q_1, q_2, ..., q_N \rangle$ and document $d = \langle d_1, d_2, ..., d_M \rangle$ where N and M are the lengths of the query and document respectively, we obtain $q' = \langle s_1, s_2, ..., s_K, q_1, q_2, ..., q_N \rangle$ and $d' = \langle s_1, s_2, ..., s_K, d_1, d_2, ..., d_M \rangle$, where K is the (same) number of special tokens prepended to both queries and documents. Instead of utilizing the entire set of token embeddings generated by BERT, we focus exclusively on the embeddings corresponding to the special tokens. This step decouples size of the tokenized string from that of the embedded representation. Thus, given a sequence of embeddings $\langle e_1, e_2, \ldots, e_{K+N} \rangle$, obtained either from $q'$ or $d'$, we select only the first $K$ embeddings.

Once the special token embeddings are extracted from BERT, they are passed through a single linear projection layer. This layer reduces the embedding dimensionality to a size suitable for the quantisation process, i.e. the number of levels. Following the projection, we apply FSQ to the embeddings, converting the continuous embeddings into quantised codes.

### 3.2 CONTRASTIVE LOSS

Our goal is to obtain discrete representations where matching query-document pairs share as many codebook tokens as possible, while the distance from other "negative" documents is maximised. Contrastive losses are a popular choice in these settings, however they require big batch sizes to minimise bias.

To train our model, we utilize a ColBERT-style loss function. Unlike traditional ColBERT model, where the similarity between query and document embeddings is measured using cosine similarity, we compute the $\ell_2$ distance between the pre-rounded FSQ representations of queries and documents. We use $\ell_2$ distance rather than cosine similarity as we want to give the maximal score to exact matches in level space because of our use of BM25 at inference while, e.g., the codebook entry $(0, 0, 0, 0, 0)$ would return a 0 cosine-distance to all other codebook entries. Moreover, cosine similarity would conflate some linearly dependant codes together. Let $Z_q$ and $Z_d$ be the sets of quantized codes for the query and document, respectively. Our contrastive loss is defined as:

$$\mathcal{L}_c = \sum_{i \in Z_q} \max_{j \in [|Z_d|]} -||z_{q_i} - z_{d_j}||_2 \tag{3}$$

To maximise batch size despite the constraints of training on limited hardware, we use a multi-vector adaptation of GradCache (Gao et al., 2021). GradCache allows us to process sub-batches sequentially while accumulating their gradients, effectively simulating a large batch size without requiring all sub-batches to be stored in memory at once.

### 3.3 AUXILIARY ENTROPY LOSS

Even though FSQ was designed to obtain high codebook utilisation without the use of auxiliary losses, we found that, with a contrastive loss, FSQ alone does not guarantee high codebook utilisation, and that instead the codebook usage collapses often at the beginning of training. To solve this problem, we propose an auxiliary entropy loss. Our goal is to maximise the usage of the codebook by maximising the codebook entropy. To do so, for each code in each query and document, we compute all distances between the contextualised prequantized representation $z$ and each codebook

vector $\zeta$ in $C$ and use them as logits to get a probability distribution:

$$P(\zeta_i|z) = \frac{e^{-||\zeta_i - z||_2}}{\sum_{\zeta_j \in C} e^{-||\zeta_j - z||_2}} \tag{4}$$

Our regularization objective would then be just the entropy of the batch-averaged probability distribution $P$, i.e. $-\sum_{\zeta_i \in C} P(\zeta_i) \log P(\zeta_i)$. However, as preliminary experiments showed, this doesn't take into account what token is actually "sampled" by the rounding procedure of FSQ. Therefore, we use the crisp, one-hot probability $P^*(\zeta_i|z)$, which is 1 if $\zeta_i = \text{argmax}_{\zeta_j} P(\zeta_j|z)$. Since the argmax operation is non-differentiable, we optimize $P^*(\zeta_i|z)$ with straight-through estimation STE (Bengio et al., 2013) , i.e., we use the gradient of $P(\zeta_i|z)$. The full loss is then:

$$\mathcal{L}_\mathcal{H} = -\sum_{\zeta_i \in C} P^*(\zeta_i) \log P^*(\zeta_i); \ \ P^*(\zeta) = \frac{1}{b}\frac{1}{t}\sum_{i=1}^{b}\sum_{j=1}^{t} P(\zeta|z_{i;j}) \tag{5}$$

### 3.4 RETRIEVAL

To perform retrieval, we embed documents in the retrieval collection using our method, and index using an inverted index data structure. We then search using TF-IDF[1] to compute a relevance score between queries and document:

$$\text{score}(q, d) = \sum_{q_i \in q} \overbrace{f(\zeta_{q_i}, d)}^{\text{TF}} \cdot \overbrace{f(\zeta_{q_i}, D)^{-1}}^{\text{IDF}} \tag{6}$$

where $f(\zeta_{q_i}, d)$ is the frequency of the discrete token in a document, and $f(\zeta_{q_i}, D)$ is the frequency of the discrete token in the full collection. This scoring methodology, however does not take into account non-exact matches. FSQ's quantization levels can be used to compute relative distances between tokens. Similarly to our contrastive objective Eq. 3, then, we use the sum of maximum negative $\ell_2$ distances to rescore the top 5000 candidates returned by search over the inverted index using the scoring in Eq. 6. This rescoring process is very computationally lightweight. Since the distances between all possible codebook entries can be precomputed and stored in a $|C| \times |C|$ matrix, the rescoring step can be performed efficiently without significantly increasing the overall inference time. This enables the model to refine the rankings of retrieved documents while still maintaining the efficiency necessary for practical deployment.

While the learned tokenisation can be used on its own, it can complement, and be complemented by, standard lexical matching. Specifically, we retrieve documents based on text-only representations and token-only representations independently and then merge the results using Reciprocal Rank Fusion (RRF) (Cormack et al., 2009). This fusion method ensures a more balanced combination of lexical and semantic retrieval signals, leading to improved results.

## 4 EXPERIMENTAL SETUP

In this section we highlight the experimental configuration for our experiments. All our code was implemented using PyTorch (Paszke et al., 2019), using the `ir-datasets` (MacAvaney et al., 2021) package to access publicly available datasets, `pyserini` for the (Lin et al., 2021) BM25 implementation and Huggingface `transformers` (Wolf et al., 2020) library for the BERT implementation. All experiments were run on a single NVIDIA A100 GPU. Our code will be open-sourced upon paper acceptance.

### 4.1 DATASETS

The MSMARCO dataset (Bajaj et al., 2018) is a large-scale benchmark commonly used in information retrieval (IR) tasks. It is derived from real-world Bing search queries and contains three main

---

[1]The `pyserini` library that we use for our retrieval experiments provides a BM25 implementation that takes into account document length and its relation with the average length in the corpus. The library also provided a mechanism to saturate the TF component. In our setup, documents are of the same length and with an extremely high number (around 98.5%) of unique tokens within each document.

components: MSMARCO Passage Ranking, MSMARCO Document Ranking, and MSMARCO Question Answering (QA). In our experiments, we focus on the Passage Ranking task, which consists of approximately 8.8 million passages and 6,980 queries for evaluation. Each query in the Passage Ranking task is paired with a relevant passage, with relevance labels provided for training. The goal is to retrieve the most relevant passages from the corpus given a query.

We will train our model in two different settings: *supervised*, where we assume the model has had some exposure to the MSMARCO training set of query-document pairs, and an *unsupervised* setting where this dataset was not used.

**Supervised Dataset Generation** During training, we apply data augmentation and use the same MSMARCO silver-queries originally introduced in Li et al. (2023). This dataset is generated using a passage-conditioned neural model trained on the original MSMARCO query-passage training pairs. We sample one query per passage yielding a training set of 8.8M queries in total. We refer to the original paper for additional details on the data generation process. We use this dataset to train our QCR architecture. Given that the original query-generation model has had access to the golden queries, we dubbed our resulting model QCR-*supervised*.

**Unsupervised Dataset Generation** We further evaluated our model in a scenario reflective of more realistic data environments, where access to traditional training queries, such as those provided by the MSMARCO dataset, were not available. This setup simulates situations where explicit query-document pairings might be absent, as is often the case in real-world datasets. To address this, we employed Gemma-2-2b (Team Gemma et al., 2024), to autonomously generate relevant queries for each document in the dataset.[2] This approach allows us to train our retrieval model effectively without relying on large-scale annotated datasets, thus showcasing the adaptability of our method to less ideal, unstructured data scenarios. We call the resulting model trained on this data as QCR-*unsupervised*.

### 4.2 POSITIONALLY-AWARE RETRIEVAL

During inference, we improve the matching accuracy of our discrete representations by incorporating positional information into each code. Specifically, we prefix each discrete code with its position in the sequence; for example, we transform the codes $c_0, c_1, c_2$ into $0\_c_0, 1\_c_1, 2\_c_2$. This modification enables sparse retrieval methods like BM25 to match tokens only if they align in both their semantic content (i.e., code ID) and their position in the sequence.

For instance, consider a query with positional codes $0\_c52, 1\_c113, 2\_c23, \ldots$ and a document with positional codes $0\_c52, 1\_c45, 2\_c113, \ldots$. In this case, the only match occurs at position 0, where both the query and the document have the code $c52$. Although the code $c113$ appears in both the query and the document, it is at different positions (position 1 in the query and position 2 in the document). Therefore, with positional codes, these are not considered a match. We empirically found that this approach leads to better results by a small margin (roughly 2% in Success100), as it leverages the model's tendency to store information at fixed positions in the code sequence.

### 4.3 HYPERPARAMETERS

We initialise the BERT transformer architecture with the weights of the TCT-ColBERT (Lin et al., 2020) model for QCR-*supervised* and we use the weights of Contriever (unsupervised) (Izacard et al., 2022) for QCR-*unsupervised*. We use a learning rate of 1e-5, an Adam (Kingma & Ba, 2017) optimiser and no weight decay, as it encouraged codebook collapse. We use a 5-dimensional codebook where each component is divided across 5 different levels, yielding a codebook with a total of $5^5 = 3125$ distinct elements. Although we experimented with multiple batch size and discrete representation lengths, our best model was trained on a schedule of increasing batch sizes and representation lengths. The batch size increased from 256 to 2048, while the representation length increased from 150 to 250. Overall, QCR was trained for 4000 gradient steps.

---

[2]We report the prompt used in Appendix A.1.

## 4.4 BASELINE

We also consider a simpler alternative method that maps sequences of text into sequences of discrete codes using $k$-means. Specifically, we employ $k$-means clustering on token-level embeddings obtained from ColBERT to assign input tokens to codes based on their cluster IDs. Note that this baseline is in part an ingredient of the ColBERT recipe, in that $k$-means is used for ANN retrieval of candidate tokens. We begin by sampling a subset of the MSMARCO dataset and encoding each document into vectors. For each document we extract the sequence of encoded token embeddings and then aggregate these embeddings across all sampled documents to form a corpus of 50M token-level embeddings. Using this corpus we train a $k$-means model with $k$ clusters, each cluster is associated with a centroid of its members. Finally, we use the clustering model to convert all documents in MSMARCO into sequences of discrete tokens: for each document, we encode it to obtaining token embeddings $\langle e_1, e_2, ..., e_{|d|} \rangle$; we then map each $\mathbf{e_i}$ to a cluster identifier $c_i = \{1, 2, \ldots, K\}$ associated with its nearest centroid according to their euclidean distance. For retrieval, we consider these cluster identifiers as regular tokens. We report results with the discrete tokens as well as with the addition of regular words. Note that for queries and documents the transformed discrete sequences have an 1-1 mapping with their respective (input) text tokens and therefore we do not consider positional matching for scoring. We report results with a $k$-means model with $K = 32000$, which produced the best results in a small hyper-parameter tuning experiment.

## 5 RESULTS

The performance of QCR was evaluated on the MSMARCO dataset. The experiments were designed to compare our approach against traditional BM25 and assess the benefits of incorporating discrete embeddings in conjunction with textual information.

In Table 1, we report several evaluation metrics, including mean reciprocal rank (MRR), normalized discounted cumulative gain (nDCG), and precision at various cutoffs for supervised QCR. Supervised QCR demonstrated superior performance over the baseline BM25 across all evaluated metrics, supporting our hypothesis that QCR is able to generate semantically rich and discrete embeddings that capture underlying textual semantics better than BM25. We observe similar results in the unsupervised setting, although with a generally lower performance. Furthermore, the optimal results were achieved through a hybrid approach that amalgamates text with discrete representations, providing an hybrid approach that benefits from the complementary strengths of discrete and traditional text embeddings, providing a more holistic representation of text for retrieval tasks.

Even though supervised dense retrieval methods trained on the same architecture (BERT-large) outperform our method, this comes at the cost of high computational and memory overhead. Moreover, we outperform the unsupervised version of Contriever when combining words and discrete codes from QCR-unsupervised. Moreover, we surpass the unsupervised version of Contriever across all reported metrics, demonstrating the potential of integrating both word-level and semantic information through learned tokens. It is important to note, however, that the unsupervised Contriever was trained on Wikipedia passages and CCNet (Wenzek et al., 2019) data using ICT and random crops, without any exposure to MSMARCO passages, whereas our approach leverages MSMARCO data and synthetic queries.

### 5.1 ANALYSIS

**How Much do Queries and Documents match?** In Figure 2 we plot the distribution of the number of tokens in MSMARCO dev queries that match a) a random document; b) a hard negative, i.e. a random document in the top-100 returned by lexical BM25; c) the ground truth document. We can see a clear correlation between the degree of relevance and the number of matches; but the in-group variance is increasing as well, so that there is a lot of diversity in the scores of ground truth document.

**Representation Length and Batch Size** During our experimentation, we observed a notable improvement in performance correlating with increases in both batch size and representation length, as can be observed in Figure 3. This trend highlights a significant opportunity for performance enhancement through the use of larger batch sizes. Training with bigger batches typically allows for

| Model | nDCG-10 | RR-10 | Succ@10 | R@100 | Succ@100 |
|---|---|---|---|---|---|
| *Supervised* | | | | | |
| DPR (Dense) | - | **0.288** | - | - | - |
| Contriever Sup (Dense) | **0.407** | - | 0.427 | **0.891** | 0.717 |
| QCR-sup. | 0.252 | 0.194 | 0.427 | 0.707 | 0.717 |
| QCR-sup w/ rescoring | 0.273 | 0.220 | 0.461 | 0.750 | 0.760 |
| QCR-sup w/ words | 0.270 | 0.216 | 0.460 | 0.749 | 0.758 |
| QCR-sup w/ words + RRF | 0.288 | 0.229 | 0.750 | 0.795 | 0.802 |
| *Unsupervised* | | | | | |
| $k$-means ($K$=32k) | 0.115 | 0.092 | 0.199 | 0.403 | 0.404 |
| $k$-means ($K$=32k) w/ words | 0.197 | 0.158 | 0.335 | 0.602 | 0.611 |
| BM25 | 0.219 | 0.176 | 0.371 | 0.658 | 0.646 |
| Contriever Unsup (Dense) | 0.206 | - | - | 0.672 | - |
| QCR-uns. | 0.128 | 0.102 | 0.221 | 0.453 | 0.462 |
| QCR-uns. w/ rescoring | 0.160 | 0.127 | 0.280 | 0.556 | 0.566 |
| QCR-uns. w/ words | 0.169 | 0.134 | 0.298 | 0.580 | 0.590 |
| QCR-uns. w/ words + RRF | **0.227** | **0.178** | **0.395** | **0.708** | **0.717** |

Table 1: We report metrics on the MSMARCO passage dev set. QCR is trained on a subset of the MSMARCO passages and queries generated to resemble those from the MSMARCO train dataset. The RRF uses a coefficient of 0.6 for the token-only results. DPR results from Oğuz et al. (2021), Contriever results from Izacard et al. (2022). Our method yields the best results in the unsupervised setting and shows promising performance in the supervised one.

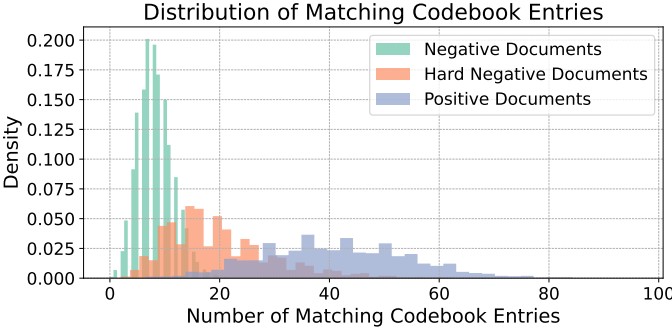

Figure 2: The plot shows the distribution of the number of matching codebook tokens between queries from the MSMARCO training set and three types of associated documents: the corresponding positive document, a hard negative document, and a randomly sampled negative document.

| Token 188 | Token 2639 |
|---|---|
| adaptations: 1.79, **pericardial**: 1.79, **receptors**: 1.79, **cysts**: 1.73, **ventricular**: 1.71, **leptin**: 1.71, **prostaglandin**: 1.71, **serotonin**: 1.70, *minerals*: 1.65, **nuclei**: 1.65, combustion: 1.58, **absorption**: 1.56, *fats*: 1.56, **cranial**: 1.55, **bloodstream**: 1.55, *diets*: 1.55, **membrane**: 1.53, sheep: 1.53, *vitamin*: 1.47, **neurons**: 1.47, encounter: 1.45, *yogurt*: 1.45, *oatmeal*: 1.44, headaches: 1.44, *supplements*: 1.42, **uric**: 1.42, sick: 1.41, *constipation*: 1.41 | **preheat**: 1.80, chartered: 1.80, **stir-fry**: 1.80, merlin: 1.80, equifax: 1.80, brat: 1.80, araya: 1.80, theodora: 1.80, dt: 1.80, cove: 1.70, ar: 1.68, **sprinkle**: 1.66, **pepper**: 1.60, **skillet**: 1.57, backup: 1.57, mp4: 1.53, **steak**: 1.49, **tablespoons**: 1.42, **seasoning**: 1.42, **sour**: 1.39, **bake**: 1.37, definitions: 1.31, **stir**: 1.30, purple: 1.28, saddle: 1.27, samsung: 1.26, **tender**: 1.25, coat: 1.25, **candy**: 1.24, irish: 1.24, **oven**: 1.22, **flour**: 1.21 |

Table 2: Selection of token-word associations, sorted using pointwise mutual information. Token 188: physiology and metabolism (bold), diet (italics). Token 2639: cooking/seasoning (bold).

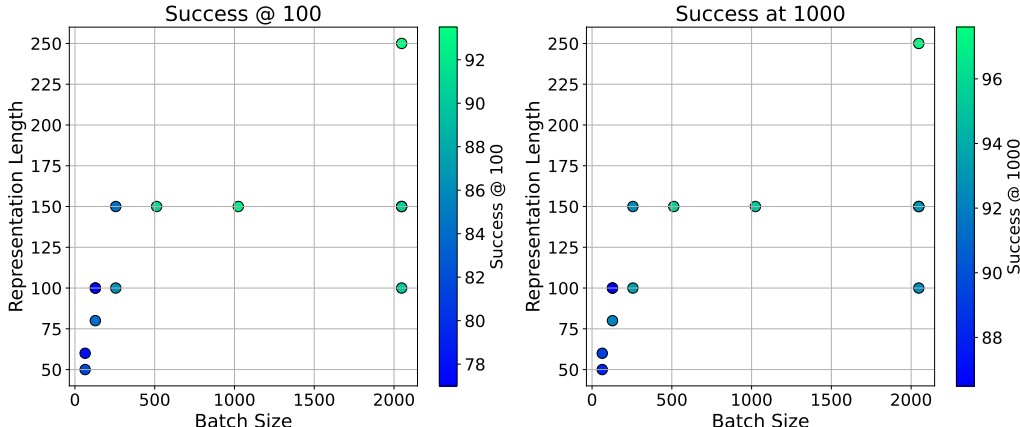

Figure 3: The plots shows the correlation between increased representation length and batch size and Success@100 metric (left) and the Success@1000 metric (right) on the hard negatives subset of the MSMARCO dataset.

more stable and accurate gradient estimates, particularly when using contrastive losses, which can lead to improved model convergence and overall effectiveness. However, due to existing hardware constraints, our experiments were limited to a maximum batch size of 2048, despite the utilization of GradCache. GradCache was instrumental in enabling us to approach this upper limit by optimizing memory usage through gradient caching techniques, yet the restriction remained a bottleneck.

**Are Latent Tokens Interpretable?** Many latent tokens predominantly associated with a topic. We include two example of this in Table 2, where we showcase latent token-natural language word associations, weighted by pointwise mutual information. Token 188 seems to be connected to a mixture of biological concepts related to health, mostly related to physiology and metabolism, and diet. Token 2639 centers around cooking, with a focus on seasoning. Not all natural language words seems to be related.

**What is the codebook usage distribution?** Our entropy loss effectively generates a uniform distribution over codebook tokens for both queries and documents, as illustrated in Figure A.2. A few rare exceptions may arise, likely due to certain topics appearing with greater frequency than others.

## 6 RELATED WORK

### 6.1 INFORMATION RETRIEVAL

In recent years, dense retrieval methods have gained prominence, driven by advances in pre-trained transformer-based models such as BERT (Devlin et al., 2019). These models generate dense vectors representations of queries and documents in a continuous embedding space, enabling retrieval based on semantic similarity rather than lexical matching alone. Dense retrieval models have demonstrated strong performance on IR tasks, especially in settings with large human-annotated datasets, as seen in works like DPR (Karpukhin et al., 2020) and ColBERT (Khattab & Zaharia, 2020).

Dense retrieval methods effectively overcome vocabulary mismatch by retrieving documents with semantically related terms, even if they differ lexically from the query. However, this advantage comes with high computational and memory costs due to large-scale matrix multiplications and the need to store dense embeddings for both queries and documents.

In response to the high resource demands of dense retrieval models, hybrid approaches have been proposed that combine dense and sparse signals to balance efficiency and effectiveness. Methods such as DeepCT (Dai & Callan, 2019), doc2query (Gospodinov et al., 2023), SPARTA (Zhao et al., 2020) and SPLADE (Formal et al., 2021) enrich sparse representations with semantic information

learned by neural networks. These hybrid models attempt to capture the strengths of both sparse and dense representations, offering improved retrieval performance while mitigating some of the computational costs associated with dense approaches. These methods are all orthogonal to QCR, and could be used in conjuction with it to improve overall performance.

## 6.2 Discrete Representation Learning

The precursor of FSQ, VQ-VAE, has shown impressive results, particularly in the image generation space, but it is notoriously challenging to optimise, with frequent codebook collapse (i.e. only a small subset of the codebook vectors are utilized by the model). Several techniques have been proposed to address the issue of codebook collapse and improve the learning of discrete representations in VQ-VAE models. One approach is the use of "random restarts" (Dhariwal et al., 2020), where underutilized codebook vectors are reset to encoder outputs, helping to ensure diverse usage. Other work, such as (Lancucki et al., 2020), improves codebook learning by periodically reinitializing the codebook using offline clustering methods.

However, preliminary experiments with both techniques in their formulation they did not succeed in preventing codebook collapse in our retrieval setting. While the goal of learning improving discretisation methods was not our primary objective, the proposed entropy loss demonstrated remarkable stability and performance in this retrieval setting. This suggests that it could serve as an effective alternative for maximizing codebook utilization in FSQ, VQ-VAE, and similar techniques. An interesting avenue for future work could be to analyse the impact of this loss on other vector quantisation methods, outside of retrieval.

## 7 Conclusion

In this paper, we introduce QCR, a system able to leverage LMs to generate discrete representations can significantly enhance the effectiveness of sparse retrieval tasks. QCR combines the semantic richness of dense embeddings with the efficiency and scalability of sparse retrieval methods, such as BM25. By introducing a novel integration of FSQ with BERT embeddings, optimized using a ColBERT-style loss, our system offers a robust approach to information retrieval that significantly outperforms traditional sparse methods in both supervised and unsupervised settings.

Furthermore, the use of these discrete representations into existing sparse retrieval infrastructures suggests a scalable and efficient pathway for enhancing the retrieval process without the need for extensive computational resources. This characteristic is particularly valuable in resource-constrained environments or scenarios where rapid scaling of information retrieval systems is required.

**Future Work** Looking ahead, the potential applications of these discrete embeddings extend beyond traditional text retrieval. QCR could be adaptable to more complex retrieval scenarios, including multimodal contexts where integration of different types of data is necessary. Future work could explore the application of discrete embeddings to help the performance of generative retrieval models, where the generative capabilities of LLMs can be harnessed to create dynamic query responses based on the rich semantic understanding encapsulated within the discrete codes. Moreover, future works might be able to dynamically control the size of the query and documents, as well optimise the distribution of the codebook to maximise throughput and minimise latency beyond what is possible for lexical term matching, which has to adapt itself to natural language statistics. Finally, training on larger models and for longer periods, with increased batch sizes, could lead to considerable performance gains, unlocking further potential of the method.

## Reproducibility Statement

Upon acceptance, we will open source our source code, alongside the hyperparameters and seed used to run the experiments. We will also be releasing the generated dataset we used in the unsupervised setting. All code used for training, evaluation, generating the dataset and baselines is included in the Supplementary Material, organized in self-contained scripts.

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

# A APPENDIX

## A.1 UNSUPERVISED DATASET GENERATION

To generate the example queries from the given documents, we use the following prompt on Gemma 2: `"You are an expert search engine query generator. Your task is to create a concise and effective query that could be used to retrieve a specific document provided by the user. The query should consist of key terms or phrases that are highly relevant to the content of the document. Your response must include only the generated query and nothing else. Document: <document>"`.

## A.2 CODEBOOK ENTROPY

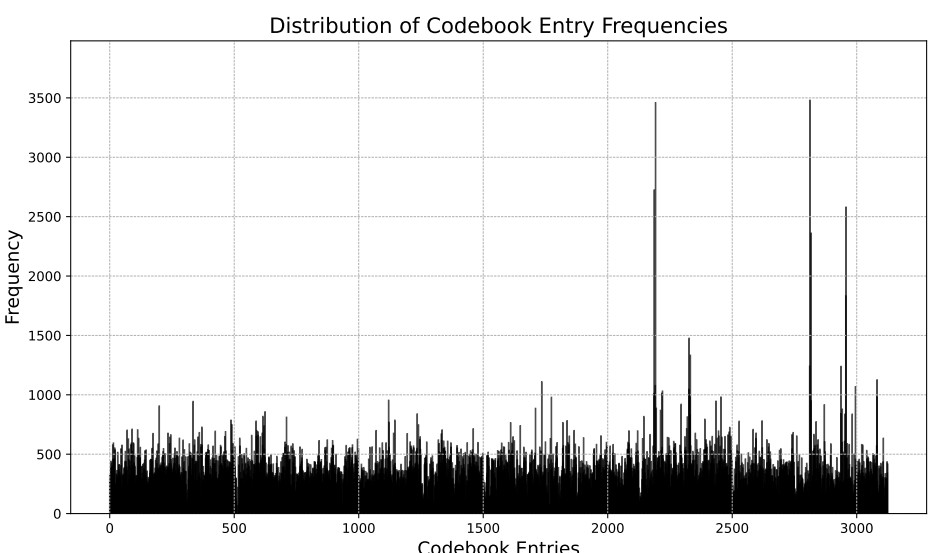

Figure 4: The plot shows the distribution of codebook tokens in the encoded MSMARCO dev set queries.

