# OpenReview forum: "QCR: Quantised Codebooks for Retrieval"
_ICLR.cc/2025/Conference — ICLR 2025 Conference Withdrawn Submission_

### Official Review · Reviewer_ruhc · 2024-10-28

**Soundness:** 2
**Presentation:** 2
**Contribution:** 1
**Rating:** 3
**Confidence:** 4

**Summary:**

This work introduces Quantised Codebooks for Retrieval (QCR), which encodes queries and documents as latent discrete tokens. QCR can replace original strings in sparse retrieval indices or complement text. Experimental results show QCR outperforms BM25 on the MSMARCO dataset, and when combined with standard lexical matching, it yields a significant gain over BM25.

**Strengths:**

(1) It is meaningful to explore discrete dense retrieval methods.

**Weaknesses:**

(1) The paper lacks a detailed explanation about the novelty or the incremental contribution, given that there are a number of studies that apply discrete representation methods for retrieval. What is the additional values?

(2) The experiments are also not convincing. There are currently a number of studies on dense retrieval, and only it is compared against several old methods.

**Questions:**

See the weakness

---

### Official Review · Reviewer_cUcR · 2024-10-28

**Soundness:** 2
**Presentation:** 2
**Contribution:** 2
**Rating:** 5
**Confidence:** 4

**Summary:**

This paper proposes a Quantised Codebooks for Retrieval (QCR) framework, which represents queries and documents using bags of latent discrete tokens. The framework employs a ColBERT-like late interaction mechanism for similarity measurement, aiming to address the vocabulary mismatch problem in traditional term-based retrieval methods like BM25. Benchmarking experiments on the MSMARCO dataset show that QCR achieves improved performance compared to the BM25 and the unsupervised Contriever.

**Strengths:**

1. This work leverages recent advancements, such as finite scalar quantization (FSQ), to learn discrete encodings for retrieval, which appears to be a novel approach.

2. This method could offer notable benefits to certain types of retrieval approaches, particularly BM25 and ColBERT, by overcoming vocabulary mismatch through discrete encodings.

**Weaknesses:**

1. The baselines are relatively weak, relying solely on outdated retrieval models like BM25 and the unsupervised Contriever. In terms of MRR@10, the performance appears low, aligning with some previous unsupervised retrieval models. Would the authors like to provide justification for this?

2. The paper overlooks a significant branch of learned sparse retrieval research[1]. The vocabulary mismatch problem is primarily relevant to BM25 and ColBERT-based methods, whereas, to my best knowledhe, learned sparse retrieval frameworks do not suffer from this issue and can still use inverted indexes, making them easier to train and retrieve compared to the approach in this work. These studies warrant citation and discussion, and the motivation should be further justified in light of this branch of research.

3. Minor: some concepts are crucial to the framework, such as "Finite Scalar Quantization (FSQ)" and "Reciprocal Rank Fusion (RRF)" with knowledge that may not be widely familiar. It would be helpful to briefly introduce these key ideas to streamline the reading experience.

[1] Formal, Thibault, et al. "SPLADE v2: Sparse lexical and expansion model for information retrieval."

**Questions:**

1. Given that the observed improvement in effectiveness is relatively modest, could the authors elaborate on the practical benefits and unique capabilities of this approach? Specifically, are there limitations in recent learned dense and sparse retrieval methods that this ColBERT-like approach (with vocabulary overlap issue addressed) uniquely overcomes?

2. The inference process appears somewhat complex; could the authors provide a cost comparison for inference relative to BM25 and the dense retrieval model Contriever?

3. The supervised version seems to underperform considerably; do the authors have any insights into the reasons for this?

**Details Of Ethics Concerns:**

To my knowledge, there is no ethics concerns in this work.

---

### Official Review · Reviewer_ixNx · 2024-11-04

**Soundness:** 2
**Presentation:** 3
**Contribution:** 2
**Rating:** 3
**Confidence:** 4

**Summary:**

The paper presents a transformer-based sparse retrieval using Finite Scalar Quantization (FSQ). The authors demonstrate that converting the embeddings from a transformer into sparse vector has the advantages of semantic (not only relying on word overlap) and lexical (supporting efficient retrieval with inverted index and generalizability) matching.

**Strengths:**

1. Although transformer-based sparse retrieval is not new, the proposed approach is interesting and this may be the first trial of FSQ on retrieval tasks.
2. The paper has good structure and easy to read.

**Weaknesses:**

The major weakness of the paper is that some claims in the introduction are not well supported.
1. The authors argue that dense retrieval underperforms traditional sparse retrieval in some demanding settings (lines 12–13), like out-of-domain; however, I do not see the authors show a fair comparison of their approach and other dense retrieval on out-of-domain tasks. Even though in the unsupervised settings, there are results on their approach and Contriever, the training data used are different. To make the claim more convincing, I think the authors should provide the dense retrieval approach fine-tuned on the same unsupervised training data (mentioned in lines 287–295). Also, are QCR-sup good at out-of-domain tasks, like BEIR[1]?
2. The authors highlight the approach’s efficiency (lines 98—100); however, the comparison of efficiency, such as query latency (the separate latency for retrieval and refining) or index storage, cannot be found in the paper.

The second part of weakness of the paper is retrieval effectiveness. Although the strength of the paper is the first trail of FSQ on retrieval tasks (as mentioned in strength	1), the paper cannot convince readers that the proposed approach is useful and effective. There are several transform-based sparse retrieval models using different codebooks, such as SPLADE[2] and UHD[3], which are shown to be more effective than the proposed approach, in terms of supervised setting, and also support inverted index retrieval.

[1] Nandan Thakur, Nils Reimers, Andreas Rücklé, Abhishek Srivastava, and Iryna Gurevych. 2021. BEIR: A heterogeneous benchmark for zero-shot evaluation of information retrieval models. In Proc. NeurIPS.
[2] Thibault Formal, Benjamin Piwowarski, and Stéphane Clinchant. 2021. SPLADE: Sparse lexical and expansion model for first stage ranking. In Proc. SIGIR. 2288–2292
[3] Kyoung-Rok Jang, Junmo Kang, Giwon Hong, Sung-Hyon Myaeng, Joohee Park, Taewon Yoon, and Heecheol Seo. 2021. Ultra-High dimensional sparse representations with binarization for efficient text retrieval. In Proc. EMNLP. 1016–1029

**Questions:**

1. Positional-aware retrieval is only performed during inference but not during training? If so, why not adding positional-aware matching mechanism into the training stage?
2. As mentioned in lines 248—250, the proposed approach can complement to heuristic lexical retrieval, such as TF-IDF or BM25. I would expect this approach can simply concat the text-only and token-only sparse vector and merge them as a unified index for retrieval. However, the authors choose to use RRF, which I assume all the other approaches, such as DPR, can be adopted. Thus, using RRF as fusion cannot reflect the advantage of the proposed method. Why the authors chose to use RRF here rather than combine heuristic and learned TF-IDF into a single inverted index for retrieval?
3. Although it is minor, I think there are many empty entries for DPR and Contriever in main experiments. The models are actually released on Huggingface and easy to run. I don’t see any reason why the authors leave those entries empty.

---

### Official Review · Reviewer_CgQA · 2024-11-04

**Soundness:** 1
**Presentation:** 3
**Contribution:** 1
**Rating:** 1
**Confidence:** 5

**Summary:**

This paper proposes a new neural late interaction IR model (QCR), loosely inspired by ColBERT. More precisely, the representation of a document or a query is given by the contextualized representations of $K$ query/document-specific tokens inserted at the beginning of a query/document. These vectors are then used as is within a ColBERT-like model (instead of the query/document tokens directly) but with a using Finite Scalar Quantization (FSQ) to discretize the vectors (eq. 3 defines the loss). The authors also propose a specific entropy-based loss to ensure good coverage of the codebook among the documents. At retrieval time, the authors propose a tf-idf-like score (eq. 6) followed by a re-ranking using eq. 3. The authors also propose to include (section 4.2) some positional information to improve somehow the results. Experiments are conducted on MS Marco (the baselines being DPR, Contriever, and BM25) and show improvement

**Strengths:**

The model itself, inspired by ColBERT, is interesting since it is a way to limit the number of vectors used to represent a document in a collection. It might even be adapted to ColBERT itself as a way to reduce the index size (but this is not investigated in the paper).

Another strength is the use of FSQ in an IR model - which could be a good option when looking at discretizing documents for faster retrieval (i.e. for dense representations).

**Weaknesses:**

The main weakness is on the experimental side.

First, the overall model does not outperform many state-of-the-art models (which are not given in the table, e.g. ColBERT, SPLADE, or better alternatives than DPR such as RetroMAE). For instance, ColBERT has an nDCG@10 of 0.47 (to be compared to 0.29 for their best setting) The authors state that these methods are "orthogonal" but I don't see how. Moreover, BM25 is already very competitive (nDCG@10 of 0.22 compared to the 0.29 of QCR), questioning the interest of the approach if speed is the main concern (although there are many alternatives in neural IR which would have a much better efficiency-effectiveness tradeoff, e.g uniCOIL, SPLADE, or TILDE).

Second, the generalization of this model is far from obvious (and no experiments were conducted on e.g. BEIR) and this is not discussed at all in the paper.

Finally, from a modeling point of view, the model in itself does not bring many novelties (using FSQ on top of a pre-trained ColBERT model as a starting point) -- and as they are not backed up by strong experiments. Even pre-pending some tokens in ColBERT has been proposed in "A Study on Token Pruning for ColBERT" (2021).

Also, there is no relationship between the score used for the first-stage retriever (eq. 6) and the loss optimized by the model. It is surprising not to exploit the discretized representation from the start (although it is not obvious how this could be done).

**Questions:**

Could you provide the full loss (eq. 3 only shows the loss for positive documents) - is it just a sum of $\mathcal L_c$  for positive documents and $-\mathcal L_c$ for negative documents?

Could you explain why you consider SOA models (l. 485) to be "orthogonal" to your approach?

I could not understand figure 3. Could you please give more details on how it was constructed?

Line 267-269, you say that you use BM25 from pyserini. However, this does not match the eq. (6). Could you clarify this?

L.315-316, you discuss an unsupervised setting - but if this is still trained with the loss in eq. 3, then this does not qualify as "unsupervised".

Could you justify the baseline (section 4.4)? In particular, what does it bring?

---

### Note · Authors · 2024-11-25

I have read and agree with the venue's withdrawal policy on behalf of myself and my co-authors.